# NMR Profiling of Exhaled Breath Condensate Defines Different Metabolic Phenotypes of Non-Cystic Fibrosis Bronchiectasis

**DOI:** 10.3390/ijms21228600

**Published:** 2020-11-14

**Authors:** Debora Paris, Letizia Palomba, Virginia Mirra, Melissa Borrelli, Adele Corcione, Francesca Santamaria, Mauro Maniscalco, Andrea Motta

**Affiliations:** 1Institute of Biomolecular Chemistry, National Research Council, 80078 Pozzuoli, Naples, Italy; debora.paris@icb.cnr.it; 2Department of Biomolecular Sciences, “Carlo Bo” University, 61029 Urbino, Italy; letizia.palomba@uniurb.it; 3Department of Translational Medical Sciences, “Federico II” University, 80131 Naples, Italy; virginia.mirra@hotmail.it (V.M.); melissaborrelli@libero.it (M.B.); dedeee@hotmail.it (A.C.); 4Pulmonary Rehabilitation Unit, ICS Maugeri SpA, IRCCS, 82037 Telese Terme (Benevento), Italy

**Keywords:** metabolomics, exhaled breath condensate, bronchiectasis, NMR, statistical analysis, biomarkers, disability, outcome

## Abstract

Nuclear-magnetic-resonance (NMR) profiling of exhaled breath condensate (EBC) provides insights into the pathophysiology of bronchiectasis by identifying specific biomarkers. We evaluated whether NMR-based metabolomics discriminates the EBC-derived metabolic phenotypes (“metabotypes”) of 41 patients with non-cystic fibrosis (nCF) bronchiectasis of various etiology [24 subjects with Primary Ciliary Dyskinesia (PCD); 17 patients with bronchiectasis not associated with PCD (nCF/nPCD)], who were compared to 17 healthy subjects (HS). NMR was used for EBC profiling, and Orthogonal Projections to Latent Structures with partial least-squares discriminant analysis (OPLS-DA) was used as a classifier. The results were validated by using the EBC from 17 PCD patients not included in the primary analysis. Different statistical models were built, which compared nCF/nPCD and HS, PCD and HS, all classes (nCF/nPCD-PCD-HS), and, finally, PCD and nCF/nPCD. In the PCD-nCF/nPCD model, four statistically significant metabolites were able to discriminate between the two groups, with only a minor reduction of the quality parameters. In particular, for nCF/nPCD, acetone/acetoin and methanol increased by 21% and 18%, respectively. In PCD patients, ethanol and lactate increased by 25% and 28%, respectively. They are all related to lung inflammation as methanol is found in the exhaled breath of lung cancer patients, acetone/acetoin produce toxic ROS that damage lung tissue in CF, and lactate is observed in acute inflammation. Interestingly, a high concentration of ethanol hampers cilia beating and can be associated with the genetic defect of PCD. Model validation with 17 PCD samples not included in the primary analysis correctly predicted all samples. Our results indicate that NMR of EBC discriminates nCF/nPCD and PCD bronchiectasis patients from HS, and patients with nCF/nPCD from those with PCD. The metabolites responsible for between-group separation identified specific metabotypes, which characterize bronchiectasis of a different etiology.

## 1. Introduction

Pediatric or adult bronchiectasis is a pathological entity associated with chronic wet cough and bronchial dilatation evident at high-resolution computed tomography (HRCT) [1]. It is universally recognized that the *primum movens* of the condition is the development of airway inflammation that persistently damages the bronchial wall, forcing the bronchi to become dilated, and hosting respiratory pathogens [2]. Bacterial colonization, associated with frequent exacerbations, fuels the vicious cycle of infection, inflammation, and bronchial damage that further increases the bacterial growth.

The etiology of bronchiectasis includes but is not restricted to, several congenital or acquired disorders including cystic fibrosis (CF), primary ciliary dyskinesia (PCD), primary immunodeficiency (PID), allergic bronchopulmonary aspergillosis, alpha-1-antitrypsin deficiency, chronic aspiration, uncontrolled asthma, chronic obstructive pulmonary disease (COPD), some rheumatologic disorders, and inflammatory bowel diseases [2]. Once ruled out, all these conditions, post-infectious bronchiectasis following recurrent pneumonia is a likely etiology in many adults or children with HRCT confirmed bronchiectasis [3,4].

Measurement of biomarkers of airway inflammation and oxidative stress in the exhaled breath condensate (EBC) is non-invasive, which makes the procedure suitable for both adults and children. Due to the close proximity between the target tissue and analytical substrate, EBC likely reflects airway inflammation [5]. Nuclear magnetic resonance (NMR)-based metabolomics of EBC has progressively gained importance as a noninvasive tool for the quantitative determination of the metabolic response in pulmonary medicine [6,7,8,9,10,11,12,13,14]. It unambiguously recognizes the metabolite markers that discriminate children or adults with airway inflammation related to asthma or COPD from healthy subjects (HS) [15,16], and patients with unstable CF from stable CF or PCD [17,18]. In a recent study on the effects of the immunomodulatory treatment in adults with non-CF bronchiectasis, we demonstrated that NMR-based metabolomics of EBC can identify changes in patients’ respiratory metabolic phenotype (or “metabotype”) [19]. In particular, by using a noninvasive approach, we showed that Pidotimod, a synthetic dipeptide molecule with biological and immunological activities, modifies the respiratory metabolic metabotype of non-CF bronchiectatic patients, which can be useful for the follow-up of bronchiectasis patients. Except for this, to our knowledge, there are no NMR-based metabolomics studies of patients with non-CF bronchiectasis. Whether the metabolomic profile of EBC from patients with non-CF bronchiectasis (i.e., the metabotype) is different is unknown as well. Herein we explored the ability of NMR-based metabolomics to characterize the EBC-derived metabotypes from patients with non-CF bronchiectasis of various etiology compared to healthy subjects. Our results, obtained from a pilot study, clearly indicate that non-CF and PCD bronchictasis patients present an EBC profile that is different from healthy subjects. Furthermore, they point out that NMR-based metabolomics efficaciously discriminates non-CF and PCD bronchiectasis, which would help in defining either the prognosis of the specific phenotype or even tailoring the individual treatment, thus contributing to improving the knowledge of the condition.

## 2. Results

Demographic and clinical characteristics and spirometry data from the enrolled subjects, including the samples used as a blind test of the statistical models are summarized in Table 1. As stated above, the 17 PCD test samples belonged to previously studied subjects [18] whose EBC samples were analyzed again in the current study.

At the time of the EBC collection, we detected the presence of pathogens in the sputum culture from patients with PCD, namely, *P. aeruginosa*, *S. aureus*, *H. influenzae*, and *S. pneumoniae* (Table 1). They were also observed in the test set.

### 2.1. NMR Profiling of EBC

Figure 1 compares the representative 1D spectra of EBC from one HS (Figure 1A), one patient with nCF/nPCD bronchiectasis (Figure 1B), and one subject with PCD (Figure 1C). Spectral resonances were assigned to single metabolites by resorting to 2D experiments (not shown) and compared with the literature data. No saliva contamination was detected, as saliva most intense signals, originating from carbohydrates resonating between 3.3 and 6.0 ppm absent in all EBC spectra.

### 2.2. Comparison of nCF/nPCD and PCD with HS

We first verified the homogeneity of the classes by applying unsupervised PCA, which is useful for outlier detection and to highlight patterns and trends. The model quality was evaluated by using the R^2^ and Q^2^ parameters. In the scores plots, no discernible patterns were identified, neither subgroups nor outliers, obtaining R^2^ = 0.17 and Q^2^ = 0.20 (*p* = 0.63) for HS (Figure 2A), R^2^ = 0.19 and R^2^ = 0.21 (*p* = 0.47) for the nCF/nPCD (Figure 2B), and R^2^ = 0.23 and Q^2^ = 0.29 (*p* = 0.58) for PCD (Figure 2C) as quality control parameters. They indicate that in all classes the absence of subgroup clustering confirms class homogeneity. Consequently, all 24, 17, and 17 samples were considered for further analysis. Similarly, the 17 PCD samples used for the external validation were proved to be homogeneous (not shown).

Next, we applied supervised OPLS-DA to HS and nCF/nPCD classes, obtaining a strong two-component model with R^2^ = 85.40% and Q^2^ = 88.82%, *p* < 0.0001. The scores plot of Figure 3A shows a well-defined nCF/nPCD cluster (red squares) for positive values of t[1], while the HS group (blue squares) is located at the negative t[1] coordinates. The chemical shifts (i.e., the spectral position of each line in an NMR spectrum) of the buckets (signals) responsible for interclass separation are reported in the associated loadings plot (Figure 3B), which shows the chemical shifts of the metabolites that discriminate the classes. The nCF/nPCD class is essentially characterized by increased levels of methanol, acetone/acetoin (signals’ overlap), ethanol, 2-propanol and propionate, and decreased concentrations of formate, acetate, lactate, and saturated fatty acids (SFAs), with acetate responsible for the HS diffusion along the p[2] axis (Table 2).

Figure 4A reports a comparison between HS and PCD classes. The OPLS-DA scores plot describes a strong two-component model with R^2^ = 84.18% and Q^2^ = 83.63%, *p* < 0.0001, showing two well-defined clusters located at positive values of t[1] (HS, blue squares), while the PCD group (green squares) is located at negative t[1] coordinates. The metabolites that discriminate the two classes are identified in the associated S-plots of Figure 4B, which shows the chemical shifts of the metabolites. In particular, the PCD class presents an increase of ethanol, methanol, lactate, and 2-propanol, and a decrease of formate, acetate, and SFAs (Table 2). When more than one chemical group belonging to the same metabolite is observed in the NMR spectrum, the corresponding buckets are all reported in the S-plot. Furthermore, if a single bucket does not cover the whole signal, two (or more) contiguous values appear in the plot. For example, for PCD, ethanol is indicated by 3.65–3.67 ppm and 1.21–1.19 ppm values, signifying that two consecutive buckets cover the CH_2_ signal resonating at 3.65 ppm, and the CH_3_ signal at 1.20 ppm. Identically, for HS, SFAs present the signals at 1.59 ppm, and two consecutive buckets (1.29–1.27 ppm) identifying the signal at 1.28 ppm.

### 2.3. All-Class Comparison

The analysis of all groups yielded a model that resulted in three predictive and three orthogonal components. In the 3D scores plot (Figure 5A), the three classes (PCD, green circles; HS, blue circles; and nFC/nPCD, red circles) are well separated along the three axes, with the following quality parameters, R^2^ = 83.32% and Q^2^ = 79.00%, *p* < 0.0001. The corresponding loadings plot, reported in 2D for clarity (Figure 5B), indicates that the nCF/nPCD class is characterized by increased levels of methanol, acetone/acetoin, and 2-propanol. The PCD group presents an increase of lactate and ethanol. Finally, the HS class is described by an increased concentration of formate, acetate, and SFAs (Table 3).

### 2.4. Comparing nCF/nPCD and PCD Classes

We also compared PCD and nCF/nPCD groups of patients, and the OPLS-DA model is reported in Figure 6A. The model was statistically significant, presenting high quality parameters, R^2^ = 87.33% and Q^2^ = 81.10%, *p* < 0.0001. The scores are well separated between PCD (green squares) and nCF/nPCD (red squares) along the t[1] axis. The associated loadings plot (Figure 6B) indicates that methanol, acetone/acetoin, 2-propanol, isobutyrate, and propionate distinguish the nCF/nPCD group. The PCD group is characterized by high levels of formate, ethanol, acetate, lactate, and SFAs (Table 4).

Considering the variable importance in projection (VIP) > 1 and *p*_corr_ > 0.6, we identified four statistically significant variables/metabolites: 1.19 ppm (ethanol), 1.33 ppm (lactate), 2.23 ppm (acetone/acetoin), and 3.37 ppm (methanol), which are labeled in the S-plot of Figure 6C. Their concentration variations are reported in Figure 7 and Figure 8. Interestingly, in nCF/nPCD, the levels of acetone/acetoin (Figure 7A) and methanol (Figure 7B) are increased by 21% and 18%, respectively, compared to PCD, while in PCD, the levels of ethanol (Figure 8A) and lactate (Figure 8B) are increased by 25% and 28%, respectively, compared to nCF/nPCD.

Then we tested whether the limited number of discriminating, statistically significant metabolites can effectively classify the PCD and nCF/nPCD groups. The statistically relevant metabolites (methanol, acetone/acetoin, lactate, and ethanol) were used for between-group classification, obtaining an 8%-reduction of the quality parameters (R^2^ = 80.34% and Q^2^ = 74.61%, *p* < 0.0001), confirming the above model. This finding indicates that a restricted number of metabolites can be used to identify some of the metabolic changes in bronchiectasis, and that they could therefore represent a functional set for discriminating PCD and nCF/nPCD samples. In particular, the levels of acetone/acetoin and methanol were confirmed to be increased in nCF/nPCD, while the levels of ethanol and lactate were confirmed to be increased in PCD. Therefore, the increase/decrease behavior in nCF/nPCD and PCD seems to suggest a pivotal role for these metabolites in the characterization and discrimination of bronchiectasis of different etiology, thus defining two different metabolic phenotypes of bronchiectasis (“metabotypes”).

### 2.5. PCD‒nCF/nPCD Model Validation

Validation of the PCD‒nCF/nPCD model is essential to prove its predictive ability for classifying unknown samples and distinguish the bronchiectasis physiopathology. This was analyzed in two steps. We firstly used an “internal” set by splitting a selected data set of 17 (out of 24 to use the same number of samples) PCD and 17 nCF/nPCD into training (67% of EBC samples) and validation (33% of EBC samples) sets for each class. Therefore, the *de-novo* PCD‒nCF/nPCD model was cyclically built with 11 PCD and 11 nCF/nPCD samples (training set), and the remaining 6 PCD and 6 nCF/nPCD samples not included in the primary analysis were used to internally validate the model. This procedure permutated all samples for a total of three runs. Classification and model authentication were achieved by using the supervised OPLS-DA approach. For each training set we obtain a robust OPLS-DA statistical model (R^2^ > 98% and Q^2^ > 97% for the I, II, and III turns). Then, we re-projected each test set onto the complementary OPLS-DA training set, which successfully classified all the test sets: 100% of PCD and nCF/nPCD correctly predicted. The PCD‒nCF/nPCD separation was due to the same markers found above, i.e., increased levels of methanol, acetone/acetoin, 2-propanol, isobutyrate, and propionate characterized the nCF/nPCD group, while higher levels of formate, ethanol, acetate, lactate, and SFAs were typically found in the PCD group.

The performance of the OPLS-DA model was also assessed using a test sample set not included in the model calculation. Specifically, we tested an external data set of EBC samples obtained from 17 PCD patients not included in the primary analysis and collected under similar experimental conditions. They originated from previously studied subjects [18] and re-examined in the current study. NMR spectra were re-acquired to verify that no differences between previous and current data (i.e., spectra wholly superimposable) existed, and only then the spectral profiles were used in the present study.

They were projected onto the corresponding statistical model obtained by using the same number of samples for each class of the training set, namely 17 nCF/nPCD and 17 (out of 24) PCD, and the results are displayed in Figure 9. The new observations were classified with respect to the model and predicted scores and responses (values of Y variables). All samples were correctly predicted (green squares with a gray rectangle in the middle), and the PCD‒nCF/nPCD separation was due to the same markers found in all the statistical models: increased methanol, acetone/acetoin, 2-propanol, isobutyrate, and propionate for nCF/nPCD, and high levels of formate, ethanol, acetate, lactate, and SFAs for PCD.

### 2.6. Pathway Topology Analysis for PCD versus nCF/nPCD Comparison

The biological significance of the data was investigated by uncovering the metabolic pathways dysregulated by the concentration alterations of the molecular species characterizing the studied classes. For this, we used the MetaboAnalyst 4.0 software. Figure 10 depicts the impact percentage of the involved pathways versus *p* values. We obtained 6 potentially affected pathways in the PCD‒nCF/nPCD comparison, which appear to be differently dysregulated. The most probable divergent metabolisms are pyruvate (*p* = 1.51 × 10^−5^; impact, 0.24), glycolysis/gluconeogenesis (*p* = 3.24 × 10^−5^; impact, 0.18) and glyoxylate and dicarboxylate metabolism (*p* = 0.0049; impact, 0.16), which are all classified in the carbohydrate metabolism.

### 2.7. Correlations

No correlation was found between biomarker data and spirometry and anthropometric parameters reported in Table 1.

## 3. Discussion

We have previously reported that stable PCD, stable CF, and healthy subjects have different EBC metabolic profiles at NMR spectroscopy. In particular, we observed that the levels of acetate, ethanol, and/or short-chain fatty acids were significantly different in the three classes of enrolled subjects, questioning that these markers are associated with airway inflammation [18].

In the current study, the clear class separation obtained from the EBC metabolic profiles identified the specific phenotypes of nCF/nPCD and PCD bronchiectasis. In particular, compared to controls, in nCF/nPCD we found an increase of methanol, acetone/acetoin, ethanol, 2-propanol, and propionate, and a decrease of formate, acetate, lactate, and SFAs levels, while subjects with PCD-associated bronchiectasis showed an increase of methanol, ethanol, 2-propanol and lactate, and a reduction of formate, acetate, and SFAs.

The comparison between nCF/nPCD and PCD uncovered a metabolic profile that seems specific for each group of patients with bronchiectasis.

For this model, we identified statistically significant metabolites, namely, ethanol and lactate (25% and 28% increase, respectively, in PCD), and acetone/acetoin and methanol (21% and 18% increase, respectively, in nFC/nPCD). This profile identifies a functional set for discriminating PCD from nCF/nPCD, and indicates that the found metabolites can differentiate the two metabotypes. Taken together, our data indicate that, although the clinical manifestations are similar, they do not generate a “molecular uniformity”, and that NMR-based metabolomics can clearly define the specificity via EBC.

The most probable divergent metabolisms between PCD and nCF/nPCD are pyruvate, glycolysis/gluconeogenesis and glyoxylate, and dicarboxylate (Figure 10). The pyruvate pathway, which includes all processes involving pyruvate, is related to energy requirement. It is central for the pathways essential in glucose and energy homeostasis because of its conversion into fatty acids, carbohydrates, and energy through acetyl-CoA and ethanol, which was increased in PCD. Pyruvic acid, produced from glucose via the tricarboxylic acid (Krebs) cycle, works as a potent scavenger of free radicals due to its antioxidant and anti-inflammatory properties [20].

In glycolysis, glucose is transformed into pyruvate and produces small quantities of ATP (which provides energy) and NADH (endowed with reducing activity), and altered glucose metabolism or glycemic control can play a role in pulmonary infectious exacerbations [21,22]. Glucose can be produced from noncarbohydrate precursors (lactate, propionate, glycerol, and the amino acids alanine and glutamine) via gluconeogenesis, an anabolic pathway. In it, lactate, which increases in PCD, is the main supply of carbon atoms for glucose production.

The glyoxylate and dicarboxylate cycle employs acetate, which is upregulated in PCD, for growth and energy [23]. The increased levels of ethanol in nCF/nPCD or formate and acetate in PCD indicate an increased energy requirement [24] and have been implicated in an extra energy demand in patients with associated obesity and asthma [8].

Many of the discriminating metabolites are related to inflammation. Methanol and formate, which increased in nCF/nPCD and PCD, respectively, are downgrading products of formaldehyde, which worsens airways inflammation in A549 alveolar and BEAS-2B bronchial cell lines [25], and in male Wistar rats [26]. Interestingly, a high level of methanol is found in the exhaled breath of patients with lung cancer [27], while formate shows an antiproliferative activity on lung cancer cell lines [27], and diminishes in the sputum from patients with lung cancer [28]. Therefore, it is tempting to hypothesize that the combined increased production of methanol in nCF/nPCD and formate in PCD points towards the activation of a protection mechanism against pulmonary inflammation in bronchiectasis.

The higher level of acetoin (3-hydroxy-2-butanone) observed in nCF/nPCD is also associated with inflammation. It is the product of the detoxication process of acetaldehyde [29], and being involved in pulmonary redox cycling, it might produce toxic ROS that damage lung tissue [30]. High acetoin levels have been found in the EBC of patients with severe deficiency of α1 antitrypsin and pulmonary emphysema [31], in the breath of CF patients, and probably it interacts with the host immune system [32]. It should be mentioned that acetoin could also be a product of the lung microbiota as it may derive from bacterial fermentation of both pathogenic and non-pathogenic bacteria [33].

Ethanol, increased in PCD, can limit the immune response of alveolar macrophages, can amplify airway leakage, generate harmful ROS, and modify the lung antioxidants/oxidants equilibrium [34]. Interestingly, a high concentration of ethanol, which may originate from soluble adenylyl cyclase inhibition, hampers cilia beating in isolated ciliary axonemes [35]. Such an impaired mechanism could be associated with the genetic defect of PCD. Increased ethanol has also been reported as a discriminant biomarker with respect to CF patients [18].

Acetone, increased in nCF/nPCD, was identified in the bronchoalveolar lavage fluid [36] and EBC [17] of CF pediatric patients presenting different inflammation levels. High acetone concentration was also observed in the EBC of stable COPD patients with respect to both non-smoking and smoking subjects with normal lung functions [37].

High 2-propanol level, observed in nCF/nPCD, is most likely the reduction product of acetone via liver alcohol dehydrogenase [38,39]. As a marker of inflammation, a high 2-propanol level was also reported for EBC of COPD [6], and as a discriminant factor for CF with respect to healthy subjects, and between stable and unstable CF [17].

Short-chain fatty acids (SCFAs, namely acetate, propionate, and butyrate), which increase in both nCF/nPCD and PCD, appear to favor leukocyte migration to the inflammation foci and control many leukocyte functions related to the production of cytokines, eicosanoids, and chemokines [40]. Since they are also implicated in the energy requirement, their alteration might reveal the close relationship between inflammation and energy production. Interestingly, fatty acids palmitate, cholesterol crystals, and ceramide (obtained from fatty acids) all trigger NLRP3 inflammasome [41]. Moreover, since in the mevalonate pathway cholesterol is assembled with acetate units, increased acetate in PCD patients could support the cholesterol synthesis, which then triggers the activation of NLRP3 inflammasome.

The action of excess propionate in nCF/nPCD could be a reduction of inflammation response as in mice treated with propionate, which showed a limited lung inflammatory response following exposure to house dust mite [42]. Although SCFAs (acetate, propionate, and butyrate) are not present in lung tissue, we hypothesize that they could activate a protective action against airway inflammation.

Elevated EBC acetate in PCD could also be a metabolic product of oral resident bacteria, such as *Streptococcus mutans* that reduces pyruvate into acetate and lactate among other end products [43]. However, this could be safely excluded as our EBC samples were not contaminated by saliva (see above).

Lactate has a key role in energy metabolism and can be considered a marker of acute inflammation [36]. The altered concentration of lactate has been reported in the progression of Alzheimer’s disease (AD) [44], also associated with an increased level of pyruvate [42]. In addition, lactate excess can bring about a noticeable rise in ROS and apoptosis in A549 cells [45].

Although the reported statistical models well describe the enrolled subjects, our study presents some limitations. First, even though the number of subjects enrolled for each class is larger than that suggested by the backward analysis, we evaluated a relatively small sample size. Nevertheless, we were able to obtain robust discrimination models and diagnostic results for the PCD and nCF/nPCD classes, identifying the dysregulation of specific metabotypes. As an overfitted model usually yields a poor performance as it amplifies minor data fluctuations, we excluded model overfitting by a permutation test. However, further validation of our data in a larger population is warranted. Second, future studies should consider the possible “time dependence” of the metabotype during aging to define the evolution of the molecular characteristics of bronchiectasis. Furthermore, the use of other biological matrices [blood, urine, saliva, BAL, etc.] should also be included. Although EBC is a valuable biomatrix to study the airway lining fluid, we understand that the use of a single biological fluid may preclude the comprehension of the complex metabolic pathways involved in the pathogenesis of this disease. Third, we should also evaluate the possible impacts of covariates, not considered here, like drugs, comorbidities, and exacerbations on the discriminating power of this EBC metabolomic approach. Finally, a lack of correlation of clinical parameters with the found metabolites may limit the interpretation of the data. This could be related to the ‘‘low resolution’’ that the use of a single biofluid can produce in the description of a complex living system, which is fully described by the clinical parameters. The study of metabolic profiles from several biofluids (serum, urine, EBC, BAL, and saliva) should better characterize the molecular bases of the pathology. Furthermore, inclusion in the molecular data of cellular and inflammatory mediators should better define cellular sources and pathophysiological mechanisms of the disease. In this way, the metabolites should be connected to the pathogenetic function of the metabolic pathways in which they play a role, most likely describing the correlation with clinical parameters.

Notwithstanding these limitations, the data from this pilot study, achieved by applying an unbiased methodology, clearly indicate that bronchiectasis of different origins presents a distinct respiratory metabolic profile and this may help clinicians in the diagnosis and tailored treatment of patients.

## 4. Materials and Methods

### 4.1. Patients

We designed a pilot, prospective, cross-sectional study. We enrolled consecutively 41 patients with non-CF bronchiectasis who attended the outpatient clinic of the Pulmonology Unit, Department of Translational Medical Sciences, Federico II University, Naples, Italy. The diagnosis of bronchiectasis had been previously made according to published criteria [4]. In all patients, CF had been previously excluded because of normal sweat chloride test and negative results at CF gene mutation analysis for the most common mutations in our population, which accounts for 94% of mutant alleles found locally. The study population was divided into two groups. Twenty-four subjects had bronchiectasis associated with PCD previously diagnosed based on abnormal cilia motility and ultrastructure, henceforth referred to as PCD (19 had *situs viscerum inversus*) [46]. Seventeen patients had bronchiectasis not associated with PCD (hereafter referred to as nCF/nPCD). In the nCF/nPCD group, 6 cases with primary immune deficiency [PID] had total IgA defect (2 cases), or iper-IgE syndrome (1 case), or Di George syndrome (1 case), or common variable immunodeficiency (2 cases). The remaining 11 subjects had developed bronchiectasis following recurrent or persistent pneumonia, not associated with any other genetic or acquired condition [3].

Inclusion criteria were: (1) evidence of bronchiectasis at HRCT; (2) ability to undergo EBC collection; and (3) adherence to the study protocol after an informed consent signature. Patients were excluded in the presence of (1) age ≤7 years old and/or technical inability to perform the EBC collection; (2) acute respiratory infection in the previous 4 weeks; (3) antibiotics assumption in the previous 4 weeks; (4) mental retardation; (5) active tobacco smoke; (6) heart disease; (7) any other respiratory chronic disorder not associated with bronchiectasis; (8) gastrointestinal disease; (9) neurologic disease. Patients had negative skin prick tests and did not report any respiratory exacerbation for which steroids, nasal decongestants, or antibiotics were administered in the previous 4 weeks.

Demographic and clinical data from patients were collected at enrollment. Spirometry and EBC collection were performed in the morning within one week since enrollment. Seventeen age- and sex-matched HS without any history of atopy or airway disorders were also enrolled for the primary analysis. All participants did not assume food or drinks (except water) during the previous 12 h.

External validation of the statistical models was performed by analyzing EBC samples from seventeen PCD patients not included in the primary analysis, which were collected following the above eligibility criteria and under similar experimental conditions. They originated from previously analyzed cases [18]. Their NMR profiles were carefully checked and superimposed to those previously collected to verify that there were no differences, and only after that, the samples were included in the current study.

The local institutional review board approved the study (Protocol n. 3/20 of Maugeri Register; 15 April 2020). Adults and children’s parents/legal guardians gave informed written consent. 

### 4.2. EBC Collection

EBC was collected by using a TURBO-DECCS condenser (Medivac, Pilastrello, Parma, Italy), and randomizing the sequential collection order, as previously reported [8]. Collection temperature was −4.8 °C ± 0.2 °C, obtaining 1.8 ± 0.3 mL (mean ± SD) of EBC for each sample. Samples, kept at −80 °C until NMR analysis, were used within a week from sampling. The room temperature was 24 °C ± 1.0 °C during collection. Salivary contamination was checked by an α-amylase activity kit (detection limit 2 U/L) (EnzCheck™ Ultra Amylase Assay Kit, Invitrogen, Paisley, UK). NMR spectra were used as a further control for saliva contamination by checking the of carbohydrates’ signals originating from saliva [16].

Air contaminants in the collecting room were ruled out because. NMR spectra of condensed room air did not show signals (data not shown).

### 4.3. NMR Spectroscopy Measurements

For NMR spectra, 70 μL of a ^2^H_2_O solution [with 1 mM sodium 3-trimethylsilyl [2,2,3,3-^2^H_4_] propionate (TSP) as internal reference for ^1^H spectra, and 3-mM sodium azide] were added to 630 μL of EBC.

NMR spectra were recorded on a 600-MHz Bruker Avance spectrometer (Bruker BioSpin GmbH, Rheinstetten, Germany) equipped with a CryoProbe™. For one-dimensional (1D) ^1^H-NMR spectra, acquired at 27 °C, we used the excitation sculpting pulse sequence to strongly reduce the intensity of the EBC water resonance [47]. Two-dimensional (2D) clean total correlation spectroscopy (TOCSY) experiments were run according to [48] and adding the excitation sculpting sequence. TSP (0.1 mM) was used as an internal 0.00-ppm reference for 1D and 2D proton spectra. 2D ^1^H-^13^C heteronuclear single quantum coherence (HSQC) spectra were acquired according to [49], and referenced to the lactate doublet (βCH_3_) resonating at 1.33 ppm for ^1^H, and 20.76 ppm for ^13^C.

### 4.4. Power Analysis

This is a pilot study for which no a priori power analysis was possible because for projection methods no standardized methods exist for evaluating the power of the analysis. From the detected biomarkers and their concentration variations, we backward assessed the power of our analysis because relevant metabolites were not known before analysis [8]. The parameters 1-α and 1-β were increased from 95% to 99.9% and from 80% to 99.9%, respectively, and using the accuracy percentages obtained in our validation tests (see Results and [8],) for a 1-α value of 95% and a 1-β value of 80%, we derived 18 ± 3 PCD, and 12 ± 2 for nCF/nPCD and HS, while for 1-α = 1-β = 99.9% we obtained 21 ± 2, 14 ± 2 and 15 ± 1, respectively. Here, for each class, we enrolled subjects complying with the numbers obtained from the backward analysis. Normally, 1-α = 95% and 1-β = 80%, and 99.9% represents is an extreme condition.

### 4.5. Demographic Statistical Analysis

Demographic data are reported as means ± SDs after checking for the normal distribution with the D’Agostino-Pearson omnibus normality test. The unpaired Student *t*-test was used to compare normally distributed values. Alternatively, the Wilcoxon–Mann–Whitney test was applied when the normality test failed. Group differences were analyzed by 1-way ANOVA, and then post-hoc multiple comparisons according to the Tukey test was applied. For each group, we carried out intraclass correlation analysis to evaluate the reliability of single measurements. Chi-square was employed to compare proportions. *p* < 0.05 was considered as statistically significant.

### 4.6. Spectral Statistical Analysis

Before spectral statistical analysis, we evaluated within-day, between-day, and technical repeatability, and detection limits according to published procedures [6,17].

For the within-day repeatability of NMR spectra, we used the Bland and Altman method as reported [17,50]. Specifically, we collected two EBC samples within the same day (at times 0 h and 4 h) for 15 subjects (5 PCD, 5 nCF/nPCD, and 5 HS). Every spectrum was divided into 6 regions (1: 8.60–6.60 ppm; 2: 6.60–5.20 ppm; 3: 4.40–3.40 ppm; 4: 3.40–2.40 ppm; 5: 2.40–1.40 ppm; 6: 1.40–0.40 ppm); the 5.20–4.50 ppm range, containing the residual water line, was excluded. Integration of all regions and normalization to the total area of the spectrum were achieved to rule out possible potential concentration variation of metabolites due to volume differences in EBC collection [17]. The 6 integrated fractional regions obtained for each spectrum gave for 15 selected subjects (spectra), 90 values in total. With an SD located in ±1.96 SD for all 90, the Bland–Altman test implies excellent within-day repeatability [50].

Between-day repeatability was assessed with the intraclass correlation coefficient (ICC). We analyzed 3 EBC samples from the above 15 subjects on days 1, 3, and 7. The 4.40–0.40 ppm spectral region was integrated and normalized to the total spectrum area. The found ICC was 0.99.

For the technical repeatability, we repeated spectra acquisition for 3 different samples (1 PCD, 1 nCF/nPCD, and 1 HS) 10 times sequentially. The ICC was 0.99.

Estimation of the detection limit was achieved normalizing the average area of 10 EBC spectra to TSP (0.1 mM) internal reference, obtaining an average concentration of 0.05 ± 0.01 μM for the endogenous phenylalanine.

For multivariate analysis, the 0.10–8.60 ppm region of NMR proton spectra were automatically data reduced to 390 integral segments (“buckets”) 0.02 ppm wide, using the AMIX v. 3.6 software package (Bruker Biospin GmbH, Rheinstetten, Germany. The 5.20–4.50 ppm region was excluded, and every region was normalized to the total spectrum area to eliminate dilution effects on the signals. The data-reduced format of the spectra was imported into SIMCA-P + v. 14 packages (Umetrics AB, Umeå/Malmö, Sweden). As signals of different intensities are present in EBC spectra, we preprocessed the data with Pareto scaling to render comparable the contribution of resonances while diminishing the effect of noise. Each region was scaled to (1/S_k_)^1/2^, with S_k_ being the standard deviation for the variable k, increasing the contribution of metabolites with lower concentration with respect to where no scaling is used. Principal Component Analysis (PCA) and Projection to Latent Structures Discriminant Analysis (PLS-DA) after orthogonal signal correction (OSC filter) of NMR data prior to chemometric analysis were also applied to reduce the influence of structured noise (spectrometer, physiological variation, etc.). We also compared Orthogonal Projections to Latent Structures (OPLS) with OSC routines together with the partial least-squares discriminant analysis (PLS-DA) to verify data fitting and possible data over-fitting. The OPLS models showed improved predictive interpretive abilities and are here reported. A permutation test (*n* = 800) was carried out to assess. A possible overfit of the model was assessed by a permutation test with *n* = 800. The goodness-of-fit (R^2^) and the goodness-of-prediction (Q^2^) parameters were used to evaluate the model quality [51].

### 4.7. Metabolic Pathway Analysis

Pathway topology and biomarker analysis on discriminating metabolites were applied to the pathway topology-search tool of MetaboAnalyst 4.0 [52]. The centrality was calculated through the Pathway Impact, a combination of the centrality and pathway enrichment results. The *Homo sapiens* pathway library was chosen and analyzed by using the Fisher exact test for overrepresentation, whereas the effect of each pathway identified was calculated by using the relative betweenness centrality [8].

## Figures and Tables

**Figure 1 ijms-21-08600-f001:**
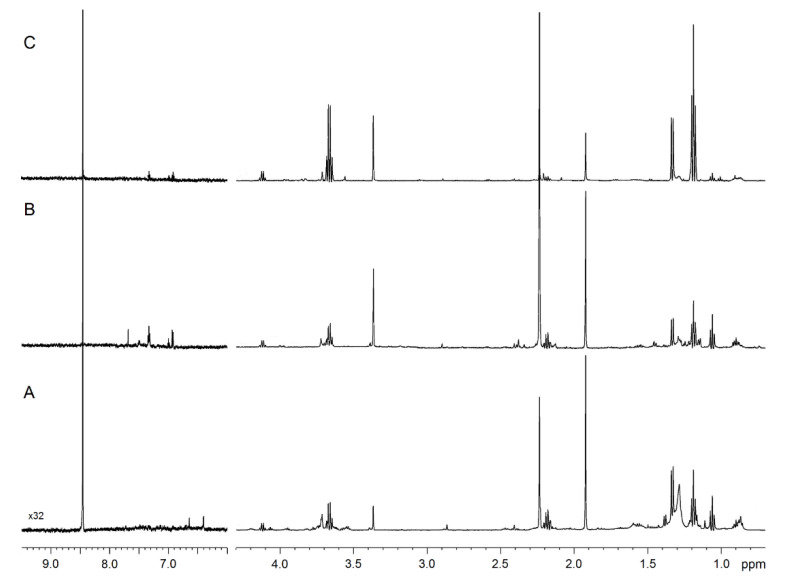
NMR spectra of exhaled breath condensate (EBC) samples. Representative 1D ^1^H spectra of a healthy subject (**A**), a non-cystic fibrosis (nCF)/non-primary ciliary dyskinesia (nPCD) bronchiectasis (**B**), and a PCD bronchiectasis (**C**). Signals were assigned to single metabolites by resorting to 2D NMR experiments and referring to published data on metabolite chemical shifts. Intensity is plotted on the y-axis, and the magnetic field strength is plotted on the x-axis that usually ranges from 0 to 12 ppm. The region between 9.5 and 6.0 ppm presents a 32-fold vertical scale increase.

**Figure 2 ijms-21-08600-f002:**
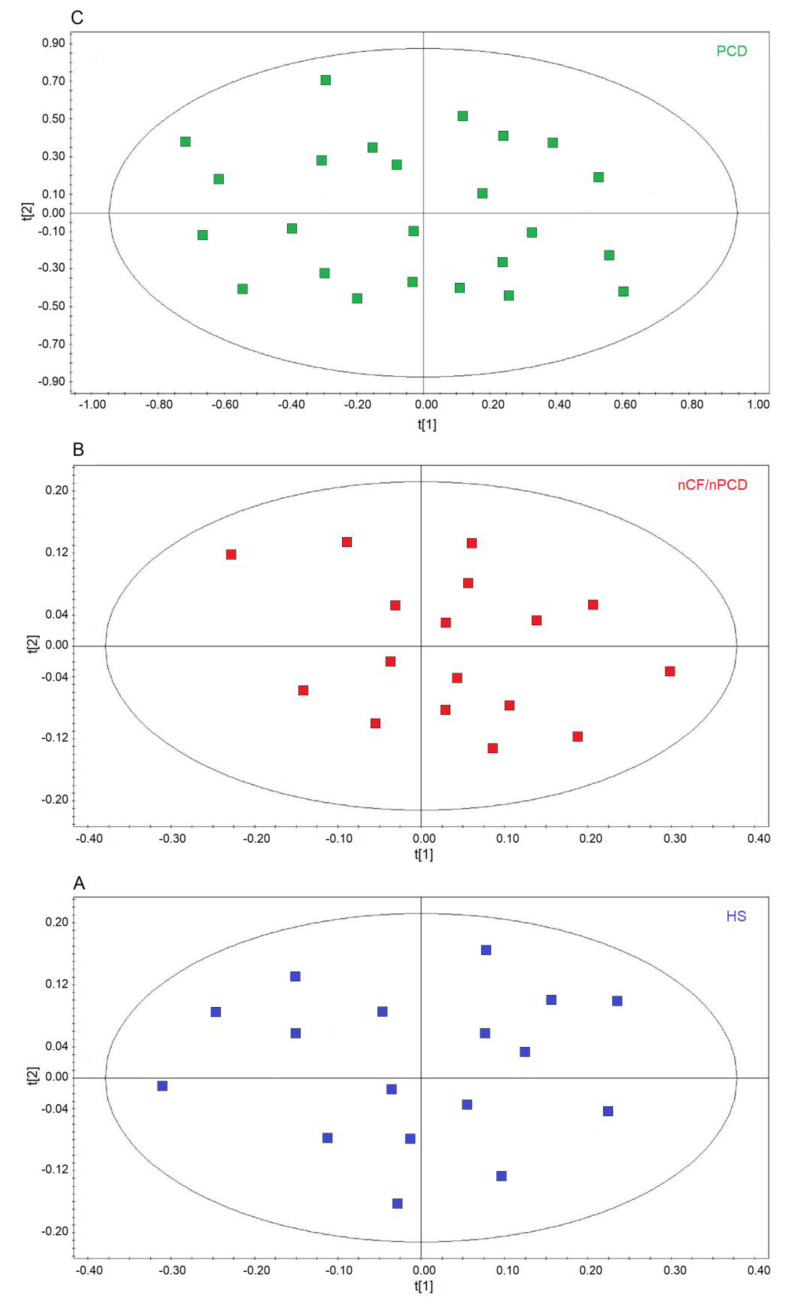
Principal component analysis (PCA) of EBC samples. (**A**) Scores plot relative to healthy subjects (HS, blue green squares); (**B**) scores plot for non-CF and non-PCD bronchiectasis patients (nCF/nPCD, red squares); (**C**) scores plot for PCD bronchiectasis patients (PCD, green squares). No satisfactory classification model was obtained, showing (**A**) R^2^ = 0.17 and Q^2^ = 0.20 (*p* = 0.63) for HS; (**B**) R^2^ = 0.19 and R^2^ = 0.21 (*p* = 0.47) for the nCF/nPCD; (**C**) R^2^ = 0.23 and Q^2^ = 0.29 (*p* = 0.58) for PCD. The labels t[1] and t[2] along the axes represent the scores (the first two partial least squares components) of the model. The absence of subgroup clustering in all plots confirms class homogeneity.

**Figure 3 ijms-21-08600-f003:**
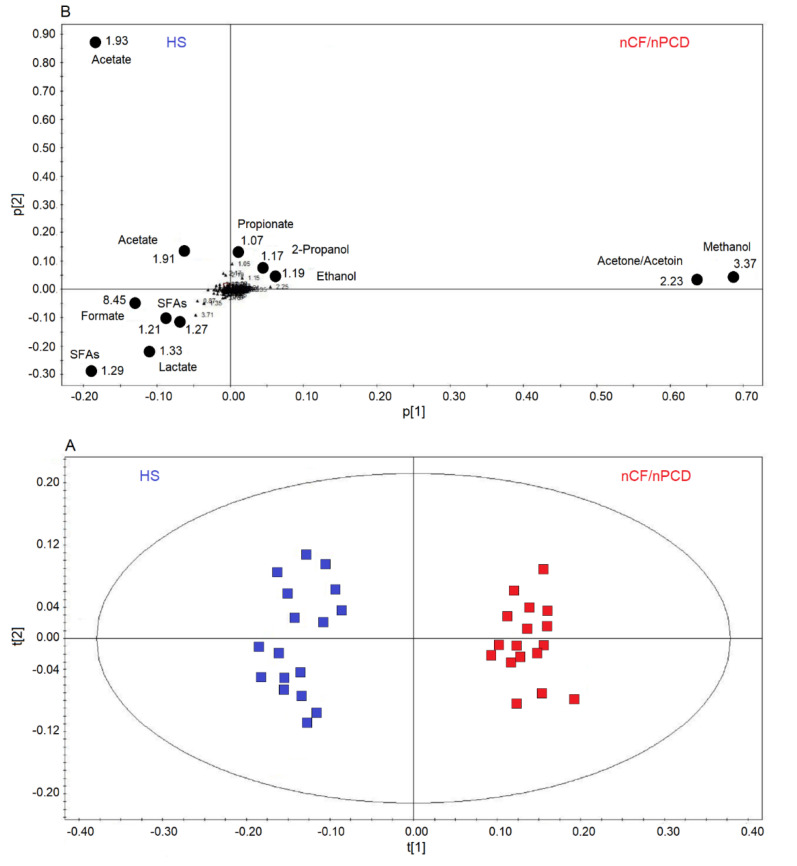
Orthogonal projections to latent structures discriminant analysis (OPLS-DA) of EBC samples for the nCF/nPCD bronchiectasis and healthy subjects (HS) comparison. (**A**) Scores plot showing the degree of separation of the model between nCF/nPCD (red squares) and HS (blue squares). The model presents strong regression (95%, *p* < 0.0001) and high-quality parameters (R^2^ = 85.40% and Q^2^ = 88.82%). (**B**) Loadings plot associated with the OPLS-DA analysis, showing the metabolites responsible for the between-class separation. Numbers refer to buckets’ chemical shifts (i.e., spectral positions), and the discriminating metabolites are explicitly labeled by black dots. The pq[1] and pq[2] values refer to the weight that combines the X and Y loadings (p and q).

**Figure 4 ijms-21-08600-f004:**
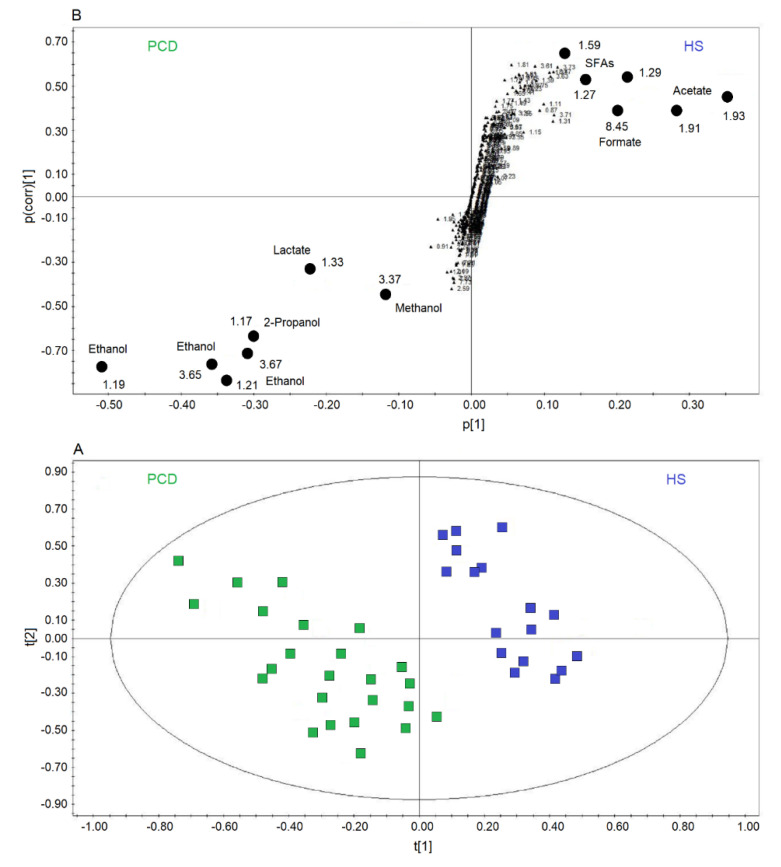
Orthogonal projections to latent structures discriminant analysis (OPLS-DA) of EBC samples for the PCD bronchiectasis and HS comparison. (**A**) Scores plot showing the degree of separation of the model between PCD (green squares) and HS (blue squares). The model presents strong regression (95%, *p* < 0.0001) and high-quality parameters (R^2^ = 84.18% and Q^2^ = 83.63%). (**B**) S-plot reporting the chemical shift of the discriminating buckets in the PCD-HS model. Numbers refer to the chemical shift of buckets, and black dots label class discriminating buckets.

**Figure 5 ijms-21-08600-f005:**
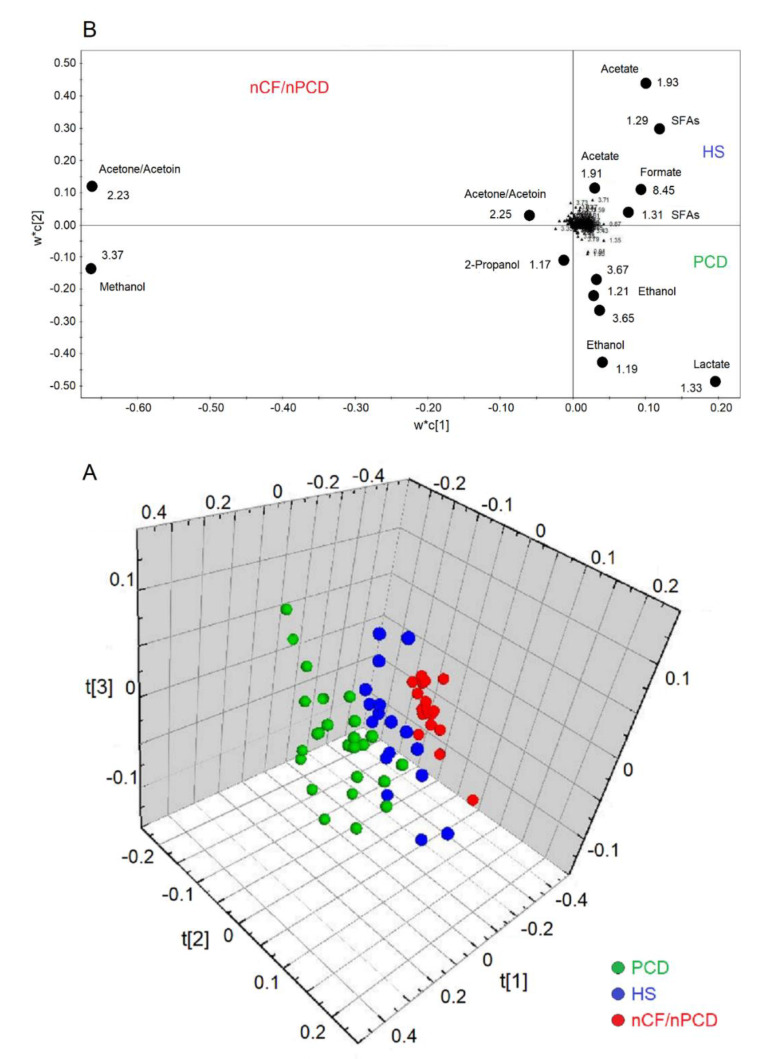
Orthogonal projections to latent structures discriminant analysis (OPLS-DA) of EBC samples for the PCD, nCF/nPCD bronchiectases, and HS comparison. (**A**) Three-dimensional scores plot showing the degree of separation of the model considering PCD (green dots) and nCF/nPCD bronchiectases (red dots), and HS (blue dots). The model presents strong regression (95%, *p* < 0.0001) with the following quality parameters, R^2^ = 83.32% and Q^2^ = 79.00%. t[1], t[2], and t[3] along the axes represent the scores (the “first three partial least squares components”) of the model, which are sufficient to build a good regression model. (**B**) Loadings plot associated with the OPLS-DA analysis reported in (**A**), showing the metabolites responsible for the between-class separation. For clarity, the loadings plot is reported in two dimensions. Numbers refer to buckets’ chemical shifts, and the discriminating metabolites (black dots) are explicitly labeled.

**Figure 6 ijms-21-08600-f006:**
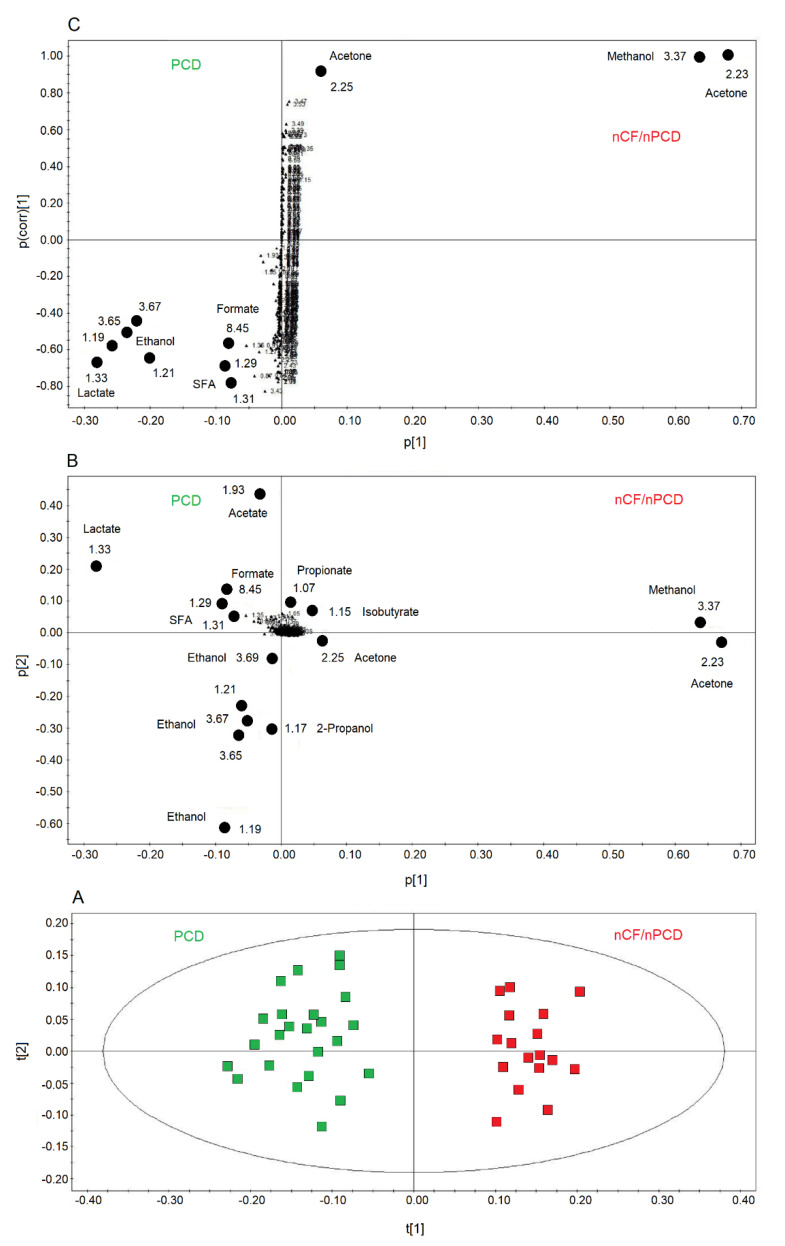
Orthogonal projections to latent structures discriminant analysis (OPLS-DA) of EBC samples for the PCD and nCF/nPCD bronchiectases comparison. (**A**) Scores plot showing the degree of separation of the model considering PCD (green squares) and nCF/nPCD bronchiectases (red squares). The model presents strong regression (95%, *p* < 0.0001) and high-quality parameters (R^2^ = 87.33% and Q^2^ = 81.10%). (**B**) Associated loadings plot showing the metabolites responsible for the between-class separation. Numbers refer to buckets’ chemical shifts, and the discriminating metabolites (black dots) are explicitly labeled. (**C**) S-plot reporting the statistically significant [Variable importance in projection (VIP) > 1 and *p*_corr_ > 0.6] discriminating buckets (black dots). Numbers refer to the chemical shift of buckets.

**Figure 7 ijms-21-08600-f007:**
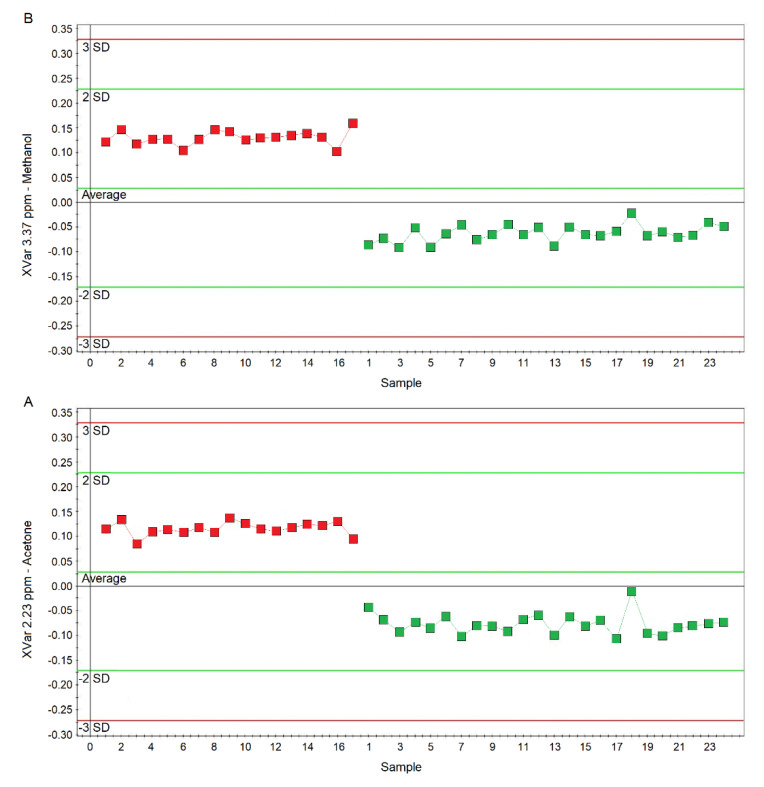
Plots of the NMR variables values for two of the four statistically significant metabolites contributing to the separation of the nPCD and nCF/nPCD groups. (**A**) Acetone/acetoin (XVar 2.23 ppm). (**B**) Methanol (XVar 3.37 ppm). Selected bucket variations are scaled to the total spectral area. nCF/nPCD patients are identified by red squares, while green squares identify PCD patients. Numbers on the x-axis refer to EBC samples used for the analysis; the y-axis reports the bucket variation. In nFC/nPCD, the levels of acetone/acetoin and methanol are increased by 21% and 18%, respectively, with respect to PCD (see text).

**Figure 8 ijms-21-08600-f008:**
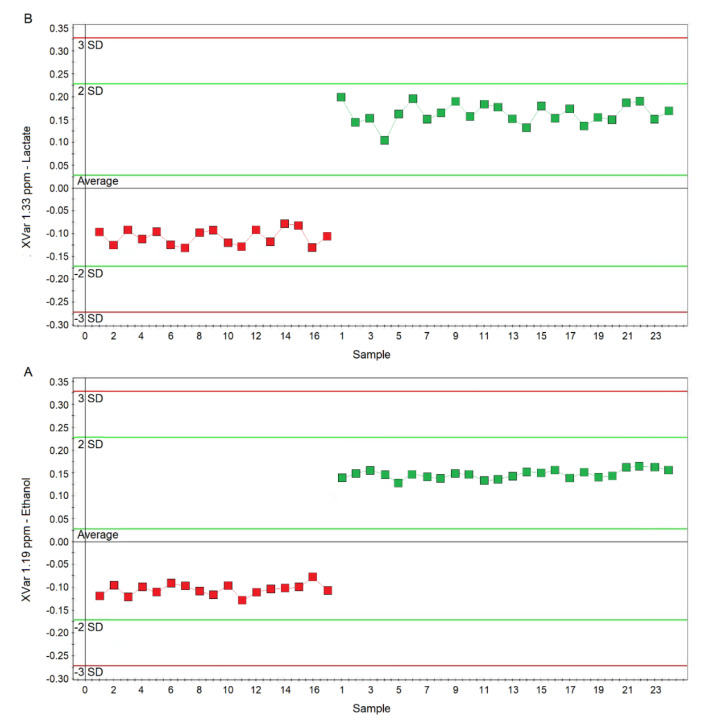
Plots of the NMR variables values for two of the four statistically significant metabolites contributing to the separation of the nPCD and nCF/nPCD classes. (**A**) Ethanol (XVar 1.19 ppm), (**B**) Lactate (XVar 1.33 ppm). Selected bucket variations are scaled to the total spectral area. nCF/nPCD patients are identified by red squares, while green squares identify PCD patients. Numbers on the x-axis refer to EBC samples used for the analysis; the y-axis reports the bucket variation. In PCD, the levels of ethanol and lactate are increased by 25% and 28%, respectively, with respect to nFC/nPCD (see text).

**Figure 9 ijms-21-08600-f009:**
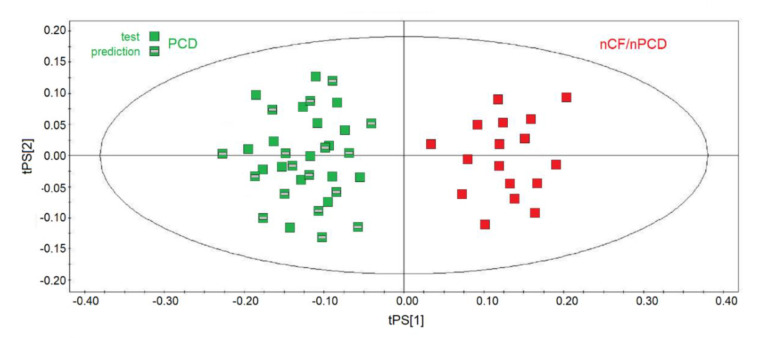
Predictive scores plot of the validation PCD set of samples. nCF/nPCD and PCD samples of the training sets are indicated by red squares and green squares, respectively, while green squares with a gray bar in the middle represent the 17-sample prediction set. The position of each sample is identified by the associated predictive coordinates (tPS).

**Figure 10 ijms-21-08600-f010:**
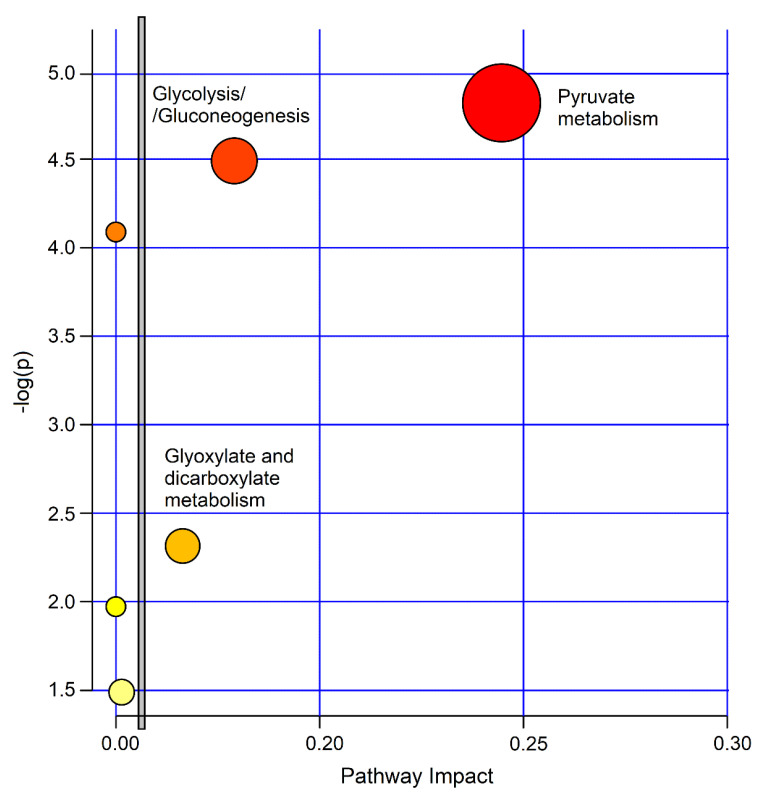
MetaboAnalyst-4.0 impact percentage of metabolic pathways as obtained from metabolites responsible for nCF/nPCD PCD class separation. Circles represent dysregulated metabolic pathways potentially involved in class separation. The most probable divergent metabolisms are labeled (pyruvate, *p* = 1.51 × 10^−5^; impact, 0.24; glycolysis/gluconeogenesis (*p* = 3.24 × 10^−5^; impact, 0.18; glyoxylate and dicarboxylate metabolism *p* = 0.00490; impact, 0.16). The vertical bar separates the 3 metabolisms with a higher impact located on the right side.

**Table 1 ijms-21-08600-t001:** Demographic, clinical, and spirometry characteristics of patients enrolled in the study ^a^.

	PCD (*n* = 24)	nCF/nPCD (*n* = 17)	HS (*n* = 17)	PCD Test Set (*n* = 17)
**Anthropometric data**				
Age, yr ^b^	17.2 ± 0.9 (7.0–33.5)	14.1 ± 1.1 (7.2–25.3)	16.8 ± 0.9 (8.1–26.9)	17.4 ± 0.9 (11.2–31.8)
Gender, M (%)/F (%)	16 (67)/8 (33)	5 (29)/12 (71)	8 (47)/9 (53)	12 (70)/5 (30)
**Chest HRCT abnormalities ^c,d^**				
Bronchiectasis, n (%)	24 (100)	17 (100)	-	17 (100)
**Sputum culture ^e^**				
*P. aeruginosa,* n (%)	1 (4)	2 (12)	-	2 (10)
*S. aureus,* n (%)	1 (4)	1 (6)	-	4 (20)
*H. influenzae,* n (%)	7 (29)	3 (18)	-	12 (60)
*S. pneumoniae,* n (%)	3 (12.5)	-	-	12 (60)
**Spirometry ^f^**				
FEV_1_, % pred	81.0 ± 20.9	83.2 ± 25.8	-	80.3 ± 2.5
FVC, % pred	90.1 ± 25.9	88.2 ± 20.4	-	92.6 ± 3.9
FEV_1_/FVC, (%)	79.0 ± 12.7	84.0 ± 17.2	-	79.9 ± 13.8
FEF_25–75_, % pred	52.3 ± 25.0	69.1 ± 39.0	-	54.1 ± 25.9

^a^ PCD, primary ciliary dyskinesia; CF, cystic fibrosis; nCF, non-cystic fibrosis; nPCD, non-primary ciliary dyskinesia; HS, healthy subjects. ^b^ Data are expressed as median and (range). ^c^ Each patient may have more than an anomaly. ^d^ Abnormalities: pneumatocele (*n* = 1); lung abscess (*n* = 1); bronchial cyst (*n* = 1). ^e^ Each culture can simultaneously be positive for multiple pathogens. ^f^ Data are expressed as mean ± SD.

**Table 2 ijms-21-08600-t002:** Discriminating metabolites for the OPLS-DA models comparing nCF/nPCD and PCD bronchiectasis with healthy subjects (see Figure 3 and Figure 4 for details). Gray arrows in the central column indicate the increase (up) or decrease (down) of metabolites in nCF/nPCD (left column) and in PCD (right column) with respect to HS ^a^.

nCF/nPCD versus HS		PCD versus HS
Methanol		Methanol
Acetone/Acetoin	‒
Ethanol	Ethanol
2-Propanol	2-Propanol
Propionate	‒
‒	Lactate
Formate	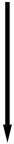	Formate
Acetate	Acetate
Lactate	‒
SFAs	SFAs

^a^ nCF, non-cystic fibrosis; nPCD, non-primary ciliary dyskinesia; HS, healthy subjects; SFAs, saturated fatty acids.

**Table 3 ijms-21-08600-t003:** Discriminating metabolites for the OPLS-DA model comparing data from healthy subjects, cases with nCF/nPCD and PCD bronchiectasis (see Figure 5 for details). Reported metabolites increase with respect to other classes ^a^.

nCF/nPCD	PCD	HS
Methanol	Lactate	Formate
Acetone/Acetoin	Ethanol	Acetate
2-Propanol		SFAs

^a^ nCF, non-cystic fibrosis; nPCD, non-primary ciliary dyskinesia; HS, healthy subjects; SFAs, saturated fatty acids.

**Table 4 ijms-21-08600-t004:** Discriminating metabolites for the OPLS-DA model comparing data from subjects with nCF/nPCD and PCD bronchiectasis (see Figure 6 for details). Reported metabolites increase with respect to the other class ^a^.

nCF/nPCD	PCD
Methanol	Formate
Acetone/Acetoin	Ethanol
2-Propanol	Acetate
Isobutyrate	Lactate
Propionate	SFAs

^a^ nCF, non-cystic fibrosis; nPCD, non-primary ciliary dyskinesia; HS, healthy subjects; SFAs, saturated fatty acids.

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
