# Peer review of "NMR Profiling of Exhaled Breath Condensate Defines Different Metabolic Phenotypes of Non-Cystic Fibrosis Bronchiectasis"

_ijms, 2020, doi:10.3390/ijms21228600_

Round 1

Reviewer 1 Report

Review of manuscript ijms-975864 entitled “NMR profiling of exhaled breath condensate defines different metabolic phenotypes of non-cystic fibrosis bronchiectasis”

SUMMARY

In this manuscript, the authors present an NMR-based study for the differentiation of EBC-derived metabotypes. The combination of NMR spectroscopy and multivariate analysis methods allowed the differentiation of the different groups and the proposal of some potential markers for each metabotype.

COMMENTARIES

In my opinion, the manuscript is well-written, and the research presented in this manuscript is sound. Most of the experimental data support the results and conclusions presented by the authors. Therefore, I believe that this manuscript could be eventually published in the International Journal of Molecular Sciences. 

However, some issues should be corrected by the authors to improve the clarity of the text and the reliability of the results.

Abstract. I think that there are too many details regarding the figures of merit of the multivariate methods and, in constrast, I miss some information related to the potential markers found in the study.

Introduction. More details regarding the novelty of this manuscript and, in particular, compared to reference 16 should be included.

Results.

I would like to see the PCA plots considering the different types of samples (Supplementary Material).

I don’t know why authors present in some cases Loadings plot and others S-plot. Please, justify.

Table 1. Do the authors think that differences in Age (significantly younger) and Gender (the unique class with more F than M) can affect the obtained results?

Discussion

Surprising metabolites (methanol, acetone, lactate and ethanol)

Methods.

Which data scaling method was used prior to the multivariate analysis?

What is the aim of the comparison of OSC+PLS-DA and OPLS-DA? Nothing is said in the results.

Author Response

  1. In my opinion, the manuscript is well-written, and the research presented in this manuscript is sound. Most of the experimental data support the results and conclusions presented by the authors. Therefore, I believe that this manuscript could be eventually published in the International Journal of Molecular Sciences.

We thank the reviewer for her/his positive comments on our manuscript.

However, some issues should be corrected by the authors to improve the clarity of the text and the reliability of the results.

  1. Abstract. I think that there are too many details regarding the figures of merit of the multivariate methods and, in constrast, I miss some information related to the potential markers found in the study.

In the revised version, we have modified the text, eliminating some numerical details of the models and added few details about the biomarkers. It nor reads: “In particular, for nCF/nPCD, …, respectively. They are all related to lung inflammation as methanol is found in the exhaled breath of lung-cancer patients, acetone/acetoin produce toxic ROS that damage lung tissue in CF, and lactate is observed in acute inflammation. Interestingly, high concentration of ethanol hampers cilia and can be associated with the genetic defect of PCD.”

(Lines 29-38).

  1. Introduction. More details regarding the novelty of this manuscript and, in particular, compared to reference 16 should be included.

We have modified the text as suggested. The new text reads:

… metabotype”) [19]. In particular, by using a noninvasive approach, we showed that Pidotimod, a synthetic dipeptide molecule with biological and immunological activities, modifies the respiratory metabolic metabotype of non-CF bronchiectatic patients, which can be useful for the follow-up of bronchiectasis patients (lines 72-76). Except for this, … to healthy subjects. Our results, obtained from a pilot study, clearly indicate that non-CF and PCD bronchictasis patients present an EBC profile that is different from healthy subjects. Furthermore, they point out that NMR-based metabolomics efficaciously discriminates non-CF and PCD bronchiectasis, which would help in defining either the prognosis of the specific phenotype or even tailoring the individual treatment, thus contributing to improve the knowledge of the condition (lines 81-86).

  1. Results.

I would like to see the PCA plots considering the different types of samples (Supplementary Material).

We have added the figure depicting the PCA plots. Since we believe that the point raised by the reviewer is relevant, we are showing the figure in the main text of the revised version. The text has been changed accordingly (lines 110 and 111).

  1. I don’t know why authors present in some cases Loadings plot and others S-plot. Please, justify.

The groupings of classes can be explained by investigating the corresponding loadings and the S plots, which are both visualization tools.

Use the loadings plot allows the identification of variables that have the largest effect on each component, while the S-plot combines the modeled covariance (X-axis) and modelled correlation (Y-axis), allowing the identification of interesting variables. The variables showing the highest p and p(corr) values are considered the most relevant metabolites for the classification between samples.

The extraction of putative biomarkers from the S-plot could be combined with the jack-knifed confidence intervals seen in the loadings plot. The advantage of the S plot is that both magnitude (intensity) and reliability is visualized, and we can obtain a list of potential biomarkers which are statistically significant. High reliability means high effect and lower uncertainty for putative biomarkers.

  1. Table 1. Do the authors think that differences in Age (significantly younger) and Gender (the unique class with more F than M) can affect the obtained results?

To account for the possible influence of age and gender as covariates on measured metabolites and remove their potential effect on NMR variables, we performed ANCOVA analysis. After means correction [i.e., factoring out (excluding) the influence of such covariates], the p-value for the selected metabolites resulted <0.05, with the difference between the metabolite levels in the classes remaining statistically significant.

  1. Surprising metabolites (methanol, acetone, lactate and ethanol)

Actually, methanol, acetone/acetoin, lactate and ethanol are the statistically significant metabolites, which were obtained from a longer list. From the biological point of view, they are not surprising. As described in the Discussion, they are involved in several processes of lung pathophysiology, which is well monitored through the EBC matrix.

  1. Methods.

Which data scaling method was used prior to the multivariate analysis?

In the revised version, we specified the scaling method, which was mistakenly omitted in the original version. The text now reads: “Since signals of different intensities are present in EBC spectra, we preprocessed the data with Pareto scaling to render comparable the contribution of resonances while diminishing the effect of noise. Each region was scaled to (1/Sk)1/2, with Sk being the standard deviation for the variable k, increasing the contribution of metabolites with lower concentration with respect to where no scaling is used. (Lines 642-646)

  1. What is the aim of the comparison of OSC+PLS-DA and OPLS-DA? Nothing is said in the results.

As said in the Materials and Methods section, the comparison was carried out to verify data fitting and rule out possible overfitting. The results indicate that “The OPLS models showed improved predictive and interpretive abilities” (line 652), and therefore OPLS was used for the analysis. For this reason, we did not compare the results, deciding to report only on the OPLS models. A thorough comparison of the results obtained with the two approaches, and, in general, on their application to EBC samples will be reported in a separate paper, which is now in progress.

Reviewer 2 Report

This paper describes an interesting and well performed NMR-based metabolomic investigation of exhaled breath condensate samples for the study of bronchiectasis patients.

I have the following minor comments:

The main drawback of the paper is the small number of samples (41 cases and 17 controls). The authors should better underline the pilot nature of the present investigation.

In Figure 2, please invert panel A and panel B, putting A on top of B, in coherence with the order in the caption.

In the introduction, when NMR-based metabolomics is introduced, these relevant publications should be cited:

https://doi.org/10.3389/fphar.2018.00258,

https://doi.org/10.1021/acs.jproteome.9b00345

https://doi.org/10.1007/s11306-013-0572-3

Author Response

  1. The main drawback of the paper is the small number of samples (41 cases and 17 controls). The authors should better underline the pilot nature of the present investigation.

The pilot nature of the study has been better underlined in the revised version (Introduction – line 81; Discussion – line 477; Materials and methods – line 485).

  1. In Figure 2, please invert panel A and panel B, putting A on top of B, in coherence with the order in the caption.

Actually, the panels in Figure 2 are correctly located. We modified slightly the text of the caption to improve the readability and avoid confusion.

  1. In the introduction, when NMR-based metabolomics is introduced, these relevant publications should be cited: https://doi.org/10.3389/fphar.2018.00258;

https://doi.org/10.1021/acs.jproteome.9b00345;

https://doi.org/10.1007/s11306-013-0572-3.

We thank the reviewer for pointing out these papers. They have been added to the revised version (references 12-14).

Finally, we have completely rewritten the text starting from the subsection 4.2 (EBC collection) to the end, which includes the 509-614 lines of the original manuscript.

Hoping the the revised manuscript fits the scientific standard of the Journal, I send my best regards.